# Role of Caspase Family in Intervertebral Disc Degeneration and Its Therapeutic Prospects

**DOI:** 10.3390/biom12081074

**Published:** 2022-08-04

**Authors:** Lei Li, Jiale He, Guangzhi Zhang, Haiwei Chen, Zhangbin Luo, Bo Deng, Yuan Zhou, Xuewen Kang

**Affiliations:** 1Department of Orthopedics, Lanzhou University Second Hospital, Lanzhou 730000, China; 2Affiliated Hospital of Xizang Minzu University, Xianyang 712082, China

**Keywords:** intervertebral disc degeneration, caspase, apoptosis, inflammation, inhibitor

## Abstract

Intervertebral disc degeneration (IVDD) is a common musculoskeletal degenerative disease worldwide, of which the main clinical manifestation is low back pain (LBP); approximately, 80% of people suffer from it in their lifetime. Currently, the pathogenesis of IVDD is unclear, and modern treatments can only alleviate its symptoms but cannot inhibit or reverse its progression. However, in recent years, targeted therapy has led to new therapeutic strategies. Cysteine-containing aspartate proteolytic enzymes (caspases) are a family of proteases present in the cytoplasm. They are evolutionarily conserved and are involved in cell growth, differentiation, and apoptotic death of eukaryotic cells. In recent years, it has been confirmed to be involved in the pathogenesis of various diseases, mainly by regulating cell apoptosis and inflammatory response. With continuous research on the pathogenesis and pathological process of IVDD, an increasing number of studies have shown that caspases are closely related to the IVDD process, especially in the intervertebral disc (IVD) cell apoptosis and inflammatory response. Therefore, herein we study the role of caspases in IVDD with respect to the structure of caspases and the related signaling pathways involved. This would help explore the strategy of regulating the activity of the caspases involved and develop caspase inhibitors to prevent and treat IVDD. The aim of this review was to identify the caspases involved in IVDD which could be potential targets for the treatment of IVDD.

## 1. Introduction

Low back pain (LBP), a common symptom of degenerative diseases of the musculoskeletal system, is one of the major diseases affecting health systems worldwide and worsens as the global population ages [1]. Due to its high morbidity and disability rates, countries around the world suffer a huge economic burden [2,3]. Intervertebral disc degeneration (IVDD) is generally believed to be an important cause of LBP [4]. Therefore, studying the pathogenesis and developing a treatment for IVDD is important.

The intervertebral disc (IVD) is the largest avascular organ in the human body [5] and consists of three distinct structures: the central gelatinous nucleus pulposus (NP), the peripheral annulus fibrosus (AF), and the upper and lower cartilaginous endplates (CEP). The central NP is surrounded by the AF and the upper and lower CEPs, forming a closed space. This closed-loop structure is crucial for maintaining the normal NP microenvironment. However, the imbalance of the NP microenvironment is often closely related to IVDD [6,7]. The main component of IVD is a highly organized extracellular matrix (ECM), which is mainly composed of proteoglycans and collagen, and is very important for maintaining the correct spine mechanics [8]. NP is a highly hydrated, viscoelastic, and gelatinous substance containing mainly type II collagen and proteoglycans. With increasing age, NP gradually decreases in water content, loses load-bearing capacity, and is not easily distinguishable from the surrounding AF tissue. AF is a concentric structure, divided into inner, middle, and outer rings, and is mainly composed of type I and type II collagen. In AF, the content of type I collagen gradually decreases from the outside to the inside, whereas the content of type II collagen gradually increases. AF mainly plays an anti-stretching role and protects the NP from damage. The CEP is a thin layer of hyaline cartilage that separates the IVD from the adjacent vertebral body. The nutrients for the IVD are mainly transported through the pores of the CEP and the outer loop of the AF. With age, the endplate becomes thinner and loses blood vessels, and the junction between the vertebral body and endplate is partially calcified, which impacts the nutrient delivery to the IVD [5,8,9,10].

Presently, the specific etiology and pathogenesis of IVDD remain unclear. It is generally believed that IVDD is caused by a variety of factors, such as genetic aspects, abnormal mechanical stress, apoptosis, increased inflammatory cytokines, and cellular senescence [11,12]. Intervertebral disc degeneration is mainly caused by an imbalance between ECM synthesis and decomposition [13]. Numerous studies have shown that pro-inflammatory factors accelerate the degradation of aggrecan and type II collagen in the ECM [12,14]. An increase in matrix degradation products triggers the production of inflammatory mediators, leading to further degradation of the ECM [15]. In addition, changes in the internal environment of the ECM lead to multiple biological changes within the IVD, thereby over-activating the IVD apoptotic pathway, which results in fewer cells within the IVD and accelerating IVDD [16]. Currently, the treatment of IVDD mainly includes conservative treatment and surgical treatment [17]. However, these treatment modalities do not reverse the progression of IVDD, but only relieve symptoms to improve the life quality of patients. Therefore, inhibition of the IVD inflammatory response and apoptosis may be a new strategy for the treatment of IVDD.

It is widely believed that IVD apoptosis and inflammatory response play important roles in the IVDD process. By studying the mechanisms of apoptosis and inflammatory response, targeting and regulating the specific mediating pathways provide hope for the treatment of IVDD [18,19]. Many studies have found that the caspase family is involved in the apoptosis and inflammatory response pathways and plays a crucial role [20]. A study on osteoarthritis (OA) found that H2O2 increased the expression of caspase-9 and -3 in chondrocytes in a concentration-dependent manner, which resulted in a decrease in chondrocyte activity and an increase in apoptosis [21]. Chen et al. [22], using the RT-PCR analysis, found that the mRNA expression levels of caspase-1 and the cytokine interleukin-1β (IL-1β) were significantly increased in degenerative IVD tissues compared to those in non-degenerative IVD tissues, which are involved in the inflammatory responses in the IVDD process. Presently, there are several in vivo and in vitro studies on the control of IVD apoptosis and inflammatory response, but few reports exist on the targeted regulation of caspases to alleviate IVDD. In this paper, with respect to the structural characteristics of the members of the caspase family and the related molecular signaling pathways involved in IVDD, the role of the caspase family in degenerated IVD tissue is described in detail, to explore the regulation of caspase activities and the development of caspase inhibitors to treat IVDD. The aim of this review was to identify caspases involved in IVDD which could be potential targets for IVDD treatment.

## 2. An Overview of Caspases

Caspases are a family of cysteine proteases homologous to the Caenorhabditis elegans programmed cell death protein, CED-3. Caspases and their distant relative’s metacaspases and paracaspases have been found in phylogenetically distant non-metazoan groups, including plants, fungi, and prokaryotes [23]. The caspases, a family of cysteine-containing aspartate proteolytic enzymes, are present in the cytoplasm. They are evolutionarily conserved and are involved in cell growth, differentiation, and apoptotic death of eukaryotic cells. There are currently 18 known mammalian-related caspases, of which 14 are related to placental mammals (Figure 1). Except for caspase-11 and -13, which are orthologs of human caspase-4 and are specifically expressed in mice and bovines, the rest are expressed in humans [24].

### 2.1. Structure

#### 2.1.1. Structure of Caspase Zymogen

The caspase zymogen, a single-chain protein with a molecular weight between 30 kDa and 50 kDa, is mainly composed of three structural regions: the amino-terminal prodomain, the large subunit region (about 20 kDa), and the carboxyl-terminal small subunit region (about 10 kDa) [25]. The large and small subunits are conserved catalytic domains; the N-terminal prodomain is located in front of the conserved domain and is involved in the recruitment and activation of proenzymes [26]. There is aspartic acid (ASP) residues between the prodomain and the large and the small subunit domains, which are usually the sites of proteolysis during caspase maturation [27]. The homology between the prodomains of zymogen is low; however, the homology between the large and small subunits is high. The prodomains of caspase family members vary widely; for example, caspases 3, 6, 7, and 14 have only short N-terminal polypeptides, whereas other caspase family members have long N-terminal domains. Among these long N-terminal domains are the death effector and caspase recruitment domains (DED and CARD) belonging to the death domain superfamily; for example, the prodomain of caspases 1, 2, 4, 5, 9, 11, and 12 contain CARD, whereas the prodomain of caspases 8 and 10 contain DED (Figure 1). These domains play an important role in the activation of caspases through protein interactions with adaptor proteins [20,28].

#### 2.1.2. Activated Caspase Structure

The activated caspase has a heterotetrameric structure composed of two large and two small subunits [29]. All caspases have a conserved pentapeptide active site: Gln-Ala-Cys-X-Gly (QACXG) (X can be R, Q or D). For example, the X position of caspase-1 is R, and the X position of caspases 8 and 10 is Q, but this does not affect the digestion of protease activity; in contrast, each pentapeptide sequence contains a cysteine residue that is necessary for the catalytic activity of the enzyme [30].

### 2.2. Activation and Maturation of Zymogen

The caspase proenzyme exists in an inactive form, and the connecting peptide between the original structure and the large and the small subunits are restrictively excised by a specific protease; thus, zymogen is converted from the inactive state to an active state, and then forms a mature tetrameric caspase. However, not all caspase activation processes are highly consistent; some caspases exist in inactive monomeric forms, such as caspases 8 and 9, relying on their long prodomains and multi-protein activation complex binding to form an active caspase by immediate dimerization. Other caspases, such as caspases 3 and 7, exist in the form of inactive dimers; with the participation of upstream caspases or other specific proteases, active caspases are formed by the limited cleavage of the interdomain structure. Active caspases further undergo proteolysis to promote caspase maturation, and mature caspases have a more stable structure and more efficient regulatory ability [27,31].

### 2.3. Substrates for Caspases

Caspases can cause the loss and gain of functions in the process of cleaving the substrate, which is related to the change in the intrinsic properties of the cleaved substrate protein [32]. For example, under normal circumstances, the caspase-activated DNA enzyme (CAD) in the nucleus combines with its molecular chaperone inhibitor of CAD (ICAD) to form a complex that inhibits CAD activity. During apoptosis, caspase-3 cleaves at D1, D2, and D3 sites of DNA fragmentation factor 45 (DFF45)/ICAD in cells. Upon cleavage at the specific sites between the three domains the complex is disrupted, the free DFF40/CAD then dimerizes to the active form and leads to DNA cleavage [33]. Similarly, the retinoblastoma tumor suppressor protein Rb is a tumor suppressant, and during tumor necrosis factor-receptor-1 (TNFR1) induced apoptosis, caspase cleaves the carboxy-terminal region of Rb, resulting in Rb degradation. The inhibitory effect on the E2F transcription factor causes the cell cycle to enter the S-phase and promotes DNA synthesis and cell proliferation [34]. These cleaved substrates lose their original functions (termed ‘loss of substrate function’). The activation of caspase itself is a type of substrate function acquisition, such as the activation of the downstream executor caspase-3 zymogen by the initiator caspase-8; thus, the apoptosis cascade reaction occurs after the activation of caspase-3 [35]. Site-specific cleavage of caspase-9 by caspase-3 abolishes the inhibitory effect of the endogenous regulator, X-linked inhibitor of apoptosis protein (XIAP) (a type of IAP), thereby promoting the apoptotic process [36]. Interestingly, in addition to the caspase family, granzyme B (GzmB) also has unique substrate specificity for cleaving aspartate residues and has been shown to activate certain caspase family members. Similarly, in terms of using non-caspase as a substrate, studies have shown that DFF45/ICAD can be cleaved by GzmB without caspase-3 activity; however, its cleavage efficiency is low, which may be related to the amplification effect of caspase on its death signal [37].

### 2.4. Classification and Functions of Caspases

The embryonic mammalian caspase family is mainly divided into inflammation-related caspases and apoptosis-related caspases according to their functions (Figure 1). Inflammation-related caspases include caspases 1, 4, 5, 11, and 12. Apoptosis-related caspases, according to their positions in the apoptosis signaling pathway, are divided into initiators (caspases 2, 8, 9, and 10) and executors (caspases 3, 6, and 7). Notably, according to recent research reports, caspase-8 is not only involved in apoptosis but also in inflammatory response processes [38,39]. Studies have shown that cigarette smoke extract (SCE) induces the TLR4-TRIF-caspase-8 signaling pathway to promote the activation of caspase-1, which causes the release of inflammatory factors [40]. Therefore, we also classified caspase-8 into an inflammation-related group. Interestingly, Koenig et al. [41] found that caspase-13 is not a human gene but bovine. Caspase-14 is mainly expressed in keratinized epithelium and plays an important role in maintaining skin homeostasis [42]. Caspases 16 and 14 have similar gene sequences [24], but their roles remain unclear. However, some caspase family members have been confirmed to be involved in the regulation of apoptosis and the inflammatory response in IVDD [43,44,45,46], which provides us with a basis to study and treat IVDD by considering the caspase family.

## 3. Link between Caspase Family and Apoptosis in IVDD

Degenerated IVD cells undergo different degrees of apoptosis, especially in the NP. The massive apoptosis of IVD cells triggers the degradation of ECM that leads to changes in homeostasis of the microenvironment, on which IVD cells depend, thereby promoting the apoptosis of IVD cells. IVD cell apoptosis is mainly induced by three pathways namely, the exogenous death receptor, endogenous mitochondrial, and endoplasmic reticulum (ER) stress pathways [47] (Figure 2). These three pathways play different roles in different degrees of IVDD. The death receptor and ER stress pathways play a major role in mild IVDD, whereas the mitochondrial pathway plays a major role in moderate and severe IVDD [48]. Caspases play an important role in these apoptotic pathways. Caspase family members induce IVD cells to undergo apoptosis by receiving and transmitting apoptotic signals, resulting in an apoptotic cascade reaction. Therefore, indirect or direct regulation of caspases can inhibit the apoptosis of IVD cells, which may result in the delay and treatment of IVDD.

### 3.1. Exogenous Death Receptor Pathway

In the exogenous death receptor pathway, death ligands combine with cell surface death receptors to induce the generation of death receptor signaling platforms, thereby triggering the initiation of apoptosis [19]. The major death receptors (DRs) involved in the pathway include: tumor necrosis factor-receptor-1 (TNFR1), Fas, and tumor necrosis factor-related apoptosis-inducing ligand receptor (TRAILR) [49]. These DRs belong to the tumor necrosis factor superfamily, which contains a unique death domain (DD) that binds to specific ligands and plays an important role in the induction of apoptosis [50].

#### 3.1.1. Tumor Necrosis Factor (TNF)/TNFR1 Pathway

The binding of TNF to the cell surface TNFR1 induces trimerization of TNFR1, and the trimerized TNFR1 recruits the TRADD through the DD. Tumor necrosis factor-receptor-associated DD recruits the Fas-associated death domain (FADD), and FADD recruits procaspase-8 through the homotypic action of its N-terminal DED, which activates caspase-8 through self-activation. Active caspase-8 cleaves downstream procaspase-3 that activates caspase-3, which subsequently promotes apoptosis by targeting substrate cleavages (Figure 2). Studies have shown that the expression of TNFR1 is significantly increased in human and mouse degenerated IVD tissues and is related to the degree of IVDD [51]. Lv et al. [52] found that the inhibition of TNFR1 attenuated LPS-mediated apoptosis of NP cells. It is well known that TNF-α is a ligand of TNFR1 and has been shown to be a key factor involved in IVDD [53]. Therefore, our research on the apoptosis of IVD cells mediated by TNF/TNFR1 is of great significance. Inhibition of caspase-8 activation in the apoptotic pathway by TNF/TNFR1 can effectively reduce apoptosis. Long et al. [54] found that activation of the JAG2/Notch2 signaling pathway could inhibit the formation of the RIP1-FADD-caspase-8 complex to inhibit TNF-α-induced apoptosis of IVD NP cells. However, the regulation of caspase-8 has always been a challenge, because it is a protease that can both promote cell apoptosis and inhibit cell necroptosis [55]. Whether the inhibition of caspase-8 activity increases the risk of necroptosis in IVD cells remains unclear. However, it is worth affirming that caspase-8 has the potential to regulate IVDD. Irigenin is an isoflavone extracted from the rhizome of She-Gan (Chinese herb) that has anti-inflammatory effects [56]. Zhang et al. [57] found that irigenin reduced the expression of caspase-3, thereby inhibiting TNF-α-induced apoptosis of NP cells. Therefore, the TNF/TNFR1 pathway involving caspase-8 and -3 plays an important role in the apoptosis of IVD cells and may become a new target for preventing and delaying IVDD in the future.

#### 3.1.2. Fas/Fas-Ligand (FasL) Pathway

FasL induces Fas receptor trimerization, and trimerized Fas recruits FADD through the homologous DD. FADD binds to the Fas receptor to recruit procaspase-8 to form a death-inducing signaling complex (DISC). Activated caspase-8 activates downstream caspase-3 to trigger the caspase apoptosis cascade and promote cell apoptosis (Figure 2). Some researchers have revealed a close association between Fas/FasL gene polymorphisms and musculoskeletal degenerative diseases through meta-analysis, among which the Fas (rs1800682) and FasL (rs763110) polymorphisms are associated with the susceptibility to IVDD [58]. In herniated lumbar IVDs, IVD cells activate the Fas/FasL signaling pathway to promote IVD cell apoptosis through autocrine or paracrine FasL [59], and Fas may be involved in apoptosis of notochord cells in NP [60]. Therefore, it may be valuable to study the Fas/FasL cell apoptotic pathway in IVDD. Inhibition of the expression of related caspases in this pathway may block the transmission of apoptotic signals and delay the process of IVDD. Wang et al. [61] found that caspase-3 is a novel target of miR-155 and that the dysregulation of miR-155 promotes Fas-mediated apoptosis of human NP cells in IVDD by targeting FADD and caspase-3. Transforming growth factor-β1 (TGF-β1) has the function of regulating cell growth and differentiation, it has a significant inhibitory effect on the activity of caspase-8 and -3 and inhibits TNF-induced apoptosis of NP cells through the Fas/FasL pathway [62]. In addition, paeoniflorin (PF), a bioactive glycoside isolated from Paeonia suffruticosa, has anti-inflammatory, antioxidant, and neuroprotective effects [63]. Chen et al. [64] found that PF could reduce the apoptosis of AF cells mediated by the Fas/FasL signaling pathway by reducing the activity of Fas and caspase-3. Although the Fas/FasL pathway has not been reported to mediate the apoptosis of CEP cells, the caspases in this pathway are closely related to the apoptosis of IVD cells; this aspect requires further research.

#### 3.1.3. Tumor Necrosis Factor-Related Apoptosis-Inducing Ligand (TRAIL)/TRAILR Pathway

TRAIL binds to TRAILR to activate apoptosis signaling, which is similar to the Fas pathway. FADD is recruited through the cognate DD, and procaspase-8 forms DISC, thereby initiating the extrinsic apoptotic pathway (Figure 2). Relevant studies have shown that the gene polymorphism of TRAIL is associated with IVDD, and the expression of TRAIL is positively correlated with the degeneration grade of IVD [65]. This is similar to the results obtained by Huang et al. [66], who assessed the association of FasL and TRAIL gene polymorphisms with IVDD risk in a meta-analysis study on 1766 IVDD cases and 1533 controls. Death receptor-4 and -5 (DR4 and DR5) are two types of TRAILR and have been shown to be important molecular mediators in the induction of apoptosis in IVD tissues [67]. Therefore, the study on the TRAIL/TRAILR axis in IVD cell apoptosis has certain value and significance. Further research revealed that some non-coding RNAs play a key role in the process of IVDD. MicroRNAs can participate in the regulation of IVD apoptosis by targeting TRAIL. Related studies have reported that TRAIL is a direct target of miR-181a and that miR-181a is down-regulated in IVDD mice, whereas TRAIL has the opposite effect. Up-regulation of miR-181a inhibits TRAIL expression and exerts anti-inflammatory and anti-apoptotic effects [68]. Xu et al. [69] found that miR-98 could target TRAIL and downregulate mRNA and protein expression levels of caspases 8 and 3 to inhibit the apoptosis of cervical disc NP cells. Therefore, we speculate that regulating the activity of related caspases through the TRAIL/TRAILR apoptosis pathway may help delay the apoptosis of IVD cells. However, few studies on this apoptotic pathway in IVD currently exist, and further research is needed to confirm our speculation.

### 3.2. Endogenous Mitochondrial Pathway

The endogenous mitochondrial apoptosis pathway is triggered by a wide range of stressors, such as DNA damage, ER stress, reactive oxygen species (ROS), and damage to mitochondrial integrity [70]. The B-cell lymphoma-2 (Bcl-2) family protein members are key regulators of anti- and pro-apoptotic effects in endogenous mitochondrial pathways. Bcl-2 family members are divided into three subfamilies according to their structure and function: Bcl-2 anti-apoptotic subfamily (Bcl-2, Bcl-xL, Bcl-w, Mcl-1, and A1), Bak pro-apoptotic subfamily (Bax, Bak, and Bok), and BH3-only protein pro-apoptotic subfamily (Bim, Bad, tBid, Bmf, Bik, Noxa, Puma, and Hrk) [71]. The aggregation of Bcl-2 family pro-apoptotic protein members in mitochondria is induced by various related stressors, resulting in a decrease in mitochondrial membrane potential and an increase in mitochondrial outer membrane permeability (MOMP). Interestingly, caspase-8 can indirectly activate the endogenous mitochondrial pathway by cleaving Bid [72], which interconnects the extrinsic and intrinsic apoptotic pathways. Pores formed during MMOP increase, allowing proteins larger than 100 kDa to cross the mitochondrial membrane into the cytoplasm [73]; mitochondria release cytochrome C (CytC) and secondary mitochondria-derived caspase activators (SMAC, DIABLO) [74]. After CytC is released into the cytoplasm, it binds to the C-terminal WD40 domain of apoptotic protease activating factor-1 (APAF1) with the assistance of dATP. The conformational changes induced by APAF1 expose the nucleotide-binding sites, which promotes the exchange of ADP for dATP; thus, accelerating the process of oligomerization of the heptameric apoptotic body structure. This structure is similar to that of a windmill. In the heptamer, APAF1 recruits procaspase-9 through the homologous CARD structure to form an apoptosome [75,76,77]. Activated caspase-9 generates a caspase cascade by activating caspases 3 and 7, cleaving the targeted proteins to promote apoptosis. XIAP is the strongest inhibitor of apoptosis among IAPs. XIAP inhibits the process of apoptosis by inhibiting the activity of caspases 3, 7, and 9 through E3 ubiquitination. However, SMAC promotes apoptosis by binding to IAPs and loses its inhibitory activity on related caspases (Figure 2).

There is considerable evidence that obesity and diabetes are associated with IVDD [78], and that the mitochondrial pathway plays a very important role in this association. Obesity induces excess gravitational pull on the spine, leading to stress-induced degeneration of the IVD. Rannou et al. [79] revealed that stress-induced apoptosis of IVD AF cells is mediated by the mitochondrial pathway and is dependent on the activation of caspase-9, rather than that of caspase-8. Bone marrow mesenchymal stem cells (BMSCs) are adult stem cells that originate from the mesoderm and have multi-directional differentiation potential. Some researchers found that in the compression-induced apoptosis of NP cells, co-culture of BMSCs and NP cells could inhibit the activation of caspases 9 and 3 and alleviate the apoptosis of NP cells by inhibiting the mitochondrial pathway [80]. Hu et al. [81] established a compression-induced apoptosis model for human NP mesenchymal stem cells (NP-MSCs). After treating the cell model with different concentrations of pioglitazone, they found that the apoptosis of NP-MSCs was alleviated upon inhibition of the endogenous mitochondrial pathway.

Diabetes mellitus (DM) is a risk factor for IVDD, activating abnormal molecular and biochemical pathways in IVD [82]. A study by Russo and colleagues showed that a murine model of type I diabetes showed signs of accelerated IDD associated with increased NP apoptosis [83]. The disease is characterized by the absorption of high glucose levels, which induces the activation of various pro-apoptotic proteins. Feng et al. [84] showed that high glucose concentrations can mediate the binding of the transcription factor carbohydrate response element binding protein (ChREBP) to p300 to promote the production of the pro-apoptotic proteins PUMA and Bax to induce the mitochondrial pathway caspase cascade and promote IVDD. Advanced glycation end products (AGEs) are known to accumulate in degenerative IVD, especially in patients with DM; however, the mechanism by which they induce apoptosis in IVD is unclear. Hu et al. [85] found that AGEs induced apoptosis in rabbit AF cells through the mitochondrial pathway. During this process, AGEs up-regulated Bax and inhibited the expression of Bcl-2; further, the activities of caspases 9 and 3 were enhanced. The apoptosis induced by IVD in a high-glucose environment can be alleviated by controlling caspases 9 and 3. Osteogenic protein-1 (OP-1) is a member of the transforming growth factor beta superfamily, which induces bone and cartilage formation and can promote the regeneration of the ECM of the IVD [86]. Liu et al. [87] found that the expression of caspase-9 and -3 was reduced when OP-1 was added to a high-glucose-induced degeneration rat NP cell model. Yao et al. [88] found that adding liraglutide, a human glucagon-like peptide-1 analogue, to human NP cells under high-glucose treatment attenuated caspase-3 by activating the PI3K/Akt/caspase-3 signaling pathway to inhibit the apoptosis of NP cells. Therefore, high glucose levels can increase the expression of caspase-3, and the regulation of caspase-3 activity may control the development of IVDD in patients with DM.

Some cellular molecules can also inhibit IVD apoptosis by modulating caspase activity in the endogenous mitochondrial pathway. Studies have shown that factors secreted by notochord cells can inhibit the apoptosis of bovine IVD NP cells by inhibiting the activity of caspases 9, 3, and 7 [89]. Sun et al. [90] found that adipose-derived stromal cells (ADSCs) could inhibit the apoptosis of human NP cells by inhibiting activated caspases 9 and 3.

In epigenetic studies, Bcl-2 family members can act as direct targets of non-coding RNAs to regulate IVDD. Some researchers have found that the silencing of miR-424-5p expression increases the expression of Bcl-2, indirectly inhibits the activation of caspases 9 and 3, and inhibits the apoptosis of NP cells. However, overexpression of miR-222 can initiate endogenous mitochondrial pathway-mediated apoptosis in NP cells by targeting Bcl-2. Similarly, overexpression of miR-573 can also target the reduction in Bax expression and inhibit the caspase apoptosis cascade generated by the activation of caspases 9 and 3, thereby inhibiting the apoptosis of NP cells [91,92,93].

Thus, we found that the endogenous mitochondrial pathway has been extensively studied in degenerated IVD tissue. Mitochondria act as a bridge linking apoptosis with ageing, autophagy, oxidative stress, inflammatory responses, and abnormal stress. The imbalance of mitochondrial membrane homeostasis induced by alterations in MMOP promotes IVD apoptosis, and the related caspases play an important role in this process. These findings indicate that inhibiting the activity of the related caspases in the endogenous mitochondrial pathway can delay and control the degenerative process of IVD.

### 3.3. Endogenous ER Stress Pathway

The ER is the processing site for the synthesis, folding, and maturation of many proteins in the cell, and plays an important role in maintaining homeostasis in the human body [94]. ER stress is caused by an imbalance in calcium ions in the lumen of the ER and the aggregation of misfolded and unfolded proteins [95]. It has been confirmed to be involved in many human diseases, such as metabolic, neurodegenerative, and genetic diseases, and cancer [96]. Unfolded protein response (UPR) signaling is activated to regulate homeostasis and promote cell survival when the accumulation of many faulty proteins in the ER reaches above a threshold level. However, when excessive ER stress occurs, the UPR of the cells promotes apoptosis [97]. This response is achieved by stimulating three transmembrane signaling molecules on the ER membrane, the inositol-requiring enzyme 1 (IRE1), activating transcription factor 6 (ATF6), and protein kinase RNA-like endoplasmic reticulum kinase (PERK) [98]. Under ER stress, the PERK and ATF6 pathways are mainly involved in apoptosis by inducing the activation of E/EBP homologous protein (CHOP). An activated CHOP affects the folding of ER proteins and causes cell cycle arrest and DNA damage, thereby triggering apoptosis [99].

Here, we mainly explore the IRE1 pathway associated with caspases. Inositol-requiring enzyme 1 has two phenotypes, IRE1α and IRE1β, and studies have shown that knockout of IRE1α in mice results in embryonic lethality [100]. IRE1 is normally associated with BIP (a stress-related molecular chaperone) to form a complex, and when unfolded proteins accumulate, IRE1 is activated after the dissociation of the complex. When ER stress persists, IRE1α suppresses survival responses and activates apoptosis through regulated IRE1-dependent decay (RIDD). Under the action of RIDD, the degradation of anti-caspase-2 microRNAs triggers the activation of the apoptotic promoter caspase-2, which subsequently triggers an endogenous mitochondria-dependent apoptotic pathway [101]. Similarly, IRE1α can also activate the ASK1-JNK signaling pathway by recruiting TRAF2, triggering apoptosis through the mitochondrial pathway, which has been demonstrated in chondrocyte apoptosis [102]. When ER stress occurs, calcium homeostasis is disrupted, and a large number of calcium ions in the ER are released into the cytoplasm through RyR and IP3R protein channels. A large influx of calcium ions in the cytoplasm causes mitochondrial permeability transition (MPT), which initiates MOMP and releases pro-apoptotic factors that induce apoptosis through the mitochondrial pathway [103].

In addition, according to a previous report, caspase-12 is involved in the process of ER stress, and caspase-12-deficient mice were found to be resistant to apoptosis induced by ER stress [104]. During ER stress, calcium release activates calpain, which activates caspase-12 by cleaving the CARD prodomain of procaspase-12. Another activation method is the release of procaspase-12 through TRAF2 recruited by IRE1α, which is automatically processed to form active caspase-12 after caspase-7 cleavage. Caspase-12 triggers mitochondria-independent induction of the caspase cascade by activating caspase-9 apoptosis [105] (Figure 2). Inhibition of IRE1 can slow down the degeneration of NP cells and prevent IVDD, which has been confirmed recently [106]; however, whether it can alleviate the exacerbation of IVDD caused by ER stress by affecting the activity of caspase-12 needs to be studied further.

## 4. Link between Caspases and Inflammatory Response in IVDD

The inflammatory response is closely related to IVDD. The inflammatory response mediated by inflammatory cytokines leads to an imbalance in IVD homeostasis, ECM degradation, and a decrease in IVD cell activity, which accelerates IVDD. Among the inflammatory cytokines involved in IVDD, TNF-α and IL-1β are the most active [107]. Numerous studies have shown that inflammatory caspases are involved in the production and release of IL-1β via the pyroptotic response pathway. Here, we mainly explore the role of caspases in inflammatory response-related pyroptosis and the effect of the related regulation on IVDD.

The pyroptosis signaling pathways are divided into canonical non-canonical pathways that depend on caspase-1 and on caspases 4, 5, and 11, respectively [108] (Figure 3). In the canonical pathway, caspase-1 is activated by promoting inflammasome assembly, and the activated caspase-1 generates mature IL-1β and IL-18 by cleaving the cytokine precursors pro-IL-1β and pro-IL-18. The assembly of the inflammasome typically consists of a sensor, the adaptor protein ASC, and procaspase-1 [109]. There are four main inflammasomes: NLRP1, NLRP3, NLRC4, and AIM2; currently, NLRP3 is the most widely studied inflammasome because it is related to the degree of IVDD [22,110,111]; inhibition of NLRP3 can block the activation of caspase-1 and thereby regulate IVD pyroptosis.

Some molecular compounds can also inhibit inflammasome formation. He et al. [112] found that in a model of human IVD NP degeneration induced by Propionibacterium acnes, MCC950 (a small molecule compound) selectively inhibited the synthesis of NLPR3, attenuated pyroptosis in human NP cells, and delayed IVDD progression. Certain molecular proteins can also regulate NLRP3; Chen et al. [113] found that a focal adhesion protein, kindlin-2, could maintain IVD homeostasis by inhibiting the activation of the NLRP3 inflammasome in NP cells.

Natural compounds also play an important role in this process. Morin is a natural flavonoid with anti-inflammatory and antioxidant effects [114]. Zhou et al. [115] found that morin alleviated the pyroptotic response in mouse disc NP cells by inhibiting the TXNIP/NLRP3/caspase-1 signaling pathway and reducing the activity of caspase-1. Honokiol, a natural bisphenol compound with anti-inflammatory, anti-aging, and anti-apoptotic properties, is mainly found in the leaves and bark of astragalus [116]. Some researchers found that in a H2O2-induced rat NP degeneration model, honokiol inhibited the generation of NLRP3 inflammasome through the TXNIP-NLRP3 signaling pathway and reduced the cleavage of pro-IL-1β by activated caspase-1 [117]. Coptisine is a quaternary ammonium alkaloid extracted from the traditional Chinese medicinal herb, Coptis chinensis, and has certain antibacterial and anti-inflammatory effects [118]. Wu et al. [119] found that coptisine can directly inhibit the active site of caspase-1 through conformational action to prevent the assembly of the NLRP3 inflammasome, which may be helpful for the treatment of gouty arthritis. Natural compounds may act on the same target in degenerated IVD cells.

Gene modification is a current research topic. METTL14 is an m6A (6-methyladenine) methyltransferase that is mainly involved in mRNA methylation modifications [120]. Yuan et al. [121] found that METTL14 is highly expressed in human degenerated NP and regulates the m6A methylation of NLRP3 mRNA, allowing insulin-like growth factor 2 mRNA-binding protein-2 (IGF2BP2) to stabilize NLRP3 mRNA and increase its protein expression. Subsequently, the delivery of miR-26a-5p to the nucleus of NP cells via exosomes from human umbilical cord mesenchymal stem cells inhibited the expression of METTL14, reduced the expression of NLRP3, and indirectly inhibited the activity of caspase-1, thereby preventing the pyroptosis in NP cells.

Certain signaling pathways also have important regulatory roles; for example, the NF-kB signaling pathway can regulate the NLRP3 inflammasome, which can effectively activate caspase-1 [122]. Song et al. [123] found that advanced glycation end products (AGEs) can promote the transcription of pro-NPLR3 and pro-IL-β in the nucleus of the NP cells by activating the NF-kB signaling pathway, while inducing ROS and calcium mobilization to trigger mitochondrial damage and lead to the NLRP3 inflammasome generation. Similarly, some studies have shown that when human NP cells are exposed to an increasing concentration of the lactic acid solution, it significantly increases the expression of acid-sensitive ion channels (ASIC1 and ASIC3) and the protein levels of NLRP3, caspase-1, and IL-1β in NP cells in a dose-dependent manner. This is because extracellular lactate can induce calcium ion influx through ASIC and promote intracellular ROS generation, thereby activating the NF-kB signaling pathway to induce pyroptosis [124]. Zhao et al. [125] found that cortistatin could inhibit the mitochondrial ROS-dependent NLRP3 inflammasome to prevent IVDD, which shows that mitochondrial-derived ROS and the NLRP3 inflammasome are closely related, which links endogenous apoptosis to pyroptosis.

In the non-canonical pathway, caspases 4, 5, and 11 undergo oligomerization after binding to lipopolysaccharide (LPS) through the cognate CARD domain to generate caspases 4, 5, and 11 with cleavage activity [126]. Similar to activated caspase-1, these active inflammatory caspases cleave the junction between the C-terminal and N-terminal domains of Gasdermin D (GSDMD), thereby releasing the N-terminal GSDMD and transporting it to the cell membrane to aggregate and assemble into pore structures. This pore-like structure can accommodate the passage of substances, such as mature IL-1β and IL-18. Thus, cleavage of GSDMD by inflammatory caspases determines the onset of pyroptosis [127]. Furthermore, in non-canonical pathways, NLRP3 inflammasome activation may be associated with elevated extracellular potassium levels. Studies have shown that caspase-11 may activate the NLRP3/ASC/caspase-1 pathway through the K+ efflux caused by GSDMD-N membrane pores, promote the release of inflammatory factors, and cause pyroptosis [128]. The regulation of GSDMD can control the release of pro-inflammatory factors, such as IL-1β, to the outside of cells to produce local inflammatory responses. The integrity of the cell membrane is crucial for the release of pro-inflammatory factors, reducing the generation of GSDMD, and increasing its degradation can prevent the pore-forming damage to the cell membrane by GSDMD-N generated by cleavage. Through in vitro experiments, Liao et al. [129] found that in an activated autophagy system, P62/SQSTM1 can selectively degrade GSDMD-N as an autophagy receptor, prevent GSDMD-N from accumulating on the cell membrane, and inhibit the pyroptosis of degenerated human NP cells. This correlates autophagy with pyroptosis and enhances the selective autophagy of pro-pyroptotic proteins. This could be a new target for the treatment of IVDD. Here we propose a similar idea; whether IVDD can be regulated by selective autophagy by caspase may be a topic worthy of more research and attention.

Thus, the NLRP3 inflammasome plays an important role in IVD cell pyroptosis in the classical pathway. Inhibiting the activity of NLRP3 inflammasome-related caspases through the regulation of artificial or natural compounds, gene modifications, and signaling pathways, is promising for the prevention and treatment of IVDD. Presently, although there are few reports on the non-canonical pathway with respect to IVDD, with our continuous in-depth research on pyroptosis, this pathway may become a research topic for the prevention and reversal of IVDD in the future.

## 5. Caspase Inhibitors

Over the years, researchers have intensively studied the involvement of the caspase family in the occurrence and development of human diseases. Indirect or direct inhibition of caspases has the potential to delay and treat certain diseases. An increasing number of studies are focused on treating diseases mediated by the caspase family by developing caspase inhibitors. Currently, caspase inhibitors are mainly divided into natural and synthetic inhibitors.

### 5.1. Natural Inhibitors

Caspase natural inhibitors mainly include IAPs, non-IAPs, and viral proteins, among others [130,131].

#### 5.1.1. Apoptotic Protein Inhibitors

Currently, there are eight types of known human-related IAPs: NAIP, XIAP, cIAP1, cIAP2, survivin, BRUCE, livin (ML-IAP, KIAP), and ILP-2 [132]; of these, XIAP may be the most effective caspase inhibitor owing to its characteristics [133]. The expression of XIAP was significantly decreased in degenerated NP tissues, resulting in up-regulation of apoptosis in NP cells. CircVMA21, as a circular RNA, can interact with miR-200c (a target gene of XIAP) and competitively target and regulate the expression of XIAP to alleviate NP cells apoptosis [134]. Survivin has been shown to be highly expressed in human degenerated NP tissues. Ma et al. [135] found that survivin could be stably expressed after transfecting degenerated human NP cells with lentiviral vector (LV)-mediated survivin in vitro, and it was found that the morphology of NP cells was significantly changed when observed under the microscope. However, apoptosis did not decrease. The team later injected the lentiviral vector survivin gene into the IVD of rabbits by in vivo puncture and found that T2-weighted images of different periods of MRI showed the water and ECM content of the intervertebral disc in the treatment group were higher than those in the puncture group. The protein content significantly increased, while caspase-3 activity and NP apoptosis significantly decreased [136], suggesting that such IAPs may have potential use for gene therapy of IVDD. In addition, the combination therapy of survivin and other genes may be able to better delay IVDD, which has been demonstrated in the experiment involving transfection of survivin-TGFB3-TIMP1 in the rabbit IVDD model [137]. Livin is a new type of IAP, and the intervention of livin contributes to the apoptosis of cancer cells, gradually becoming an attractive target for cancer therapy [138,139]; however, it has not been validated for IVDD treatment.

#### 5.1.2. Non-IAP Apoptosis Inhibitors

Non-IAP apoptosis inhibitors mainly include Bcl-2 family members and heat shock proteins [131]. Bcl-2 is located upstream of related caspases in endogenous apoptosis, and studies have shown that the -938C > A polymorphism of Bcl-2 may be significantly associated with the severity of human lumbar disc degeneration (LDD) [140]. The reduction of mitochondrial pathway-related caspase expression to alleviate IVDD by regulating Bcl-2 has been mentioned above. Interestingly, heat shock proteins (HSPs), as a non-IAP apoptosis inhibitor, can promote the expression of Bcl-2 and synergistically resist apoptosis [141]. HSPs are generally considered to be protective proteins of the body. Takao et al. [142] found that both HSP72 and HSP27 were present in the nuclei of NP, AF, and CEP by immunostaining IVD tissues from 135 cadavers and that their levels decreased with age. Furthermore, their immunoreactivity is enhanced in CEP degeneration; thus, it is speculated that HSP27 and HSP72 have a protective effect on IVDD. HSPA8 is a 70 kDa heat shock protein. Liu et al. [143] found that HSPA8 was stably expressed in normal NP tissues, but its expression gradually decreased with the increase in IVDD. However, not all HSPs are protective against IVDD, and studies have shown that the inhibition of HSP90 can prevent compression-induced NP stem/progenitor cell death [144]. We hypothesized that increasing the expression of some HSPs would treat IVDD. Although this hypothesis has been demonstrated in the treatment of OA [145], it has not been reported in the literature for the treatment of IVDD.

#### 5.1.3. Viral Proteins

Viral proteins caspase inhibitors mainly include CrmA and P35 proteins. CrmA was the first caspase inhibitor discovered in a vaccinia virus. After CrmA was fused to the TAT protein transduction domain, it showed the potential to inhibit apoptosis in mice both in vivo and in vitro [146]. HA/CS nanoparticles are very good gene carriers that can transport the CrmA gene into synovial cells. The HA/CS/pCrmA nanoparticle significantly reduced the expression level of MMP genes and relieved inflammation of synovial cells, which may have some value in the treatment of OA [147]. The P35 protein is derived from baculovirus, which itself has anti-apoptotic and antioxidant properties, and has certain therapeutic effects in neurodegenerative diseases, cancer, inflammatory arthritis, and other fields [148]. In the future, the delivery of viral protein caspase inhibitors through gene carriers may have certain potential in the treatment of IVDD, but the actual effect in the clinical application needs to be considered.

Although these natural caspase inhibitors are highly safe, they lack specificity and may be unstable during long-term use, rendering it difficult to obtain the expected therapeutic effect. In addition, it has been confirmed that in animals, natural caspase inhibitors have good therapeutic prospects; however, relevant clinical evidence is lacking, necessitating clinical translational research in the future. The effects of some natural caspase inhibitors on IVDD under the regulation of non-coding RNAs are listed in Table 1; the studies mentioned can provide a reference for subsequent research on this topic. Therefore, the development of perfect natural inhibitors may be a topic worthy of attention for the treatment of IVDD in the future.

### 5.2. Synthetic Inhibitors

Caspase synthetic inhibitors mainly include peptide and peptidomimetic inhibitors [158].

#### 5.2.1. Peptide Caspase Inhibitors

Z-VAD-FMK and Q-VD-OPh are two common synthetic peptide pan-caspase inhibitors which play a role in bone and joint diseases. Transcriptomic analysis of caspase-inhibited primary chondroblasts revealed that these two inhibitors significantly up- or down-regulated the expression of genes associated with OA development [159]. Z-VAD-FMK significantly reduced chondrocyte apoptosis in an inducible experimental rabbit OA model [160]. In IVDD, Chen et al. [161] found that the use of Z-VAD-FMK blocked compression-induced apoptosis in rat NP cells but aggravated necroptosis, whereas the combined use of the necroptosis inhibitor Necrostatin-1 promoted the survival of NP cells. This also reveals the lack of specificity of this class of pan-caspase inhibitors.

#### 5.2.2. Peptidomimetic Caspase Inhibitors

Belnacasan (VX-765) and pralnacasan (VX-740) are two peptidomimetic selective caspase-1 inhibitors that have been used in human clinical trials. These two inhibitors have been used in clinical studies in rheumatoid arthritis and OA [162]. In IVDD, delivery of VX-765 inhibits inflammasome activation and pyroptosis, thereby improving IVDD progression in vivo [129]. Although VX-765 has a good therapeutic effect on many diseases, it has certain toxic side effects and needs to be further improved for clinical applications.

The design and synthesis of many caspase inhibitors have been studied for the treatment of metabolic, neurodegenerative, and inflammatory diseases and cancer. However, owing to the general deficiencies of caspase inhibitors, specifically, poor specificity, unstable efficacy, and related side effects, there is still a long way to go for clinical treatment [163]. Due to this reason, we found that whether it is the study of natural inhibitors or synthetic inhibitors, especially those with high selectivity, people strive to maximize the therapeutic advantages of caspase inhibitors while reducing the adverse effects of inhibitors on disease. It is undeniable that caspase inhibitors have shown their therapeutic advantages and therapeutic prospects in many diseases, but little research has been conducted on their use in the treatment of IVDD. We look forward to the development of more efficient and stable inhibitors that can be applied in the clinical research on IVDD, which is very promising for the treatment of IVDD.

## 6. Conclusions and Future Perspectives

IVDD is usually the initial step in the evolution and development of a series of degenerative spinal diseases, and this degenerative process is inseparable from the apoptosis and the inflammatory response of IVD cells. Caspase is a specific proteolytic enzyme that is stably expressed in the human body and participates in cell apoptosis and inflammatory response depending on its unique properties and has been shown to be important in degenerated IVD cells. Through the elaboration of the structural characteristics of the caspase family and the role of the caspase family in the process of apoptosis and inflammatory response signaling pathways, we showed that the regulation of the caspase family is of great significance of the study on IVDD. The significance is as follows: (1) it can link apoptosis and inflammatory response and regulate the expression of related caspases to delay the progression of IVDD; (2) the target of caspase is clear and easy to regulate. However, the downside is that caspases are in a downstream position and are easily regulated by a variety of pro-producing and pro-inhibitory signals. Therefore, indirect regulation of degenerative IVD may make it difficult to achieve the desired therapeutic effect. In addition, in morphologically normal cells, the expression of caspases can prevent the accumulation of abnormal cells, which has a positive effect on maintaining cell homeostasis. However, it is difficult to precisely regulate the concentration of caspases in degenerated IVD cells to eliminate abnormal IVD cells without damaging normal cells.

Further research on the development of caspase-selective inhibitors and non-selective inhibitors and their combination may have good prospects for the treatment of IVDD. However, before caspase-targeted therapy can be applied to the clinical treatment of IVDD, controlled preclinical experimental studies and preliminary human experiments need to be performed to evaluate its therapeutic value. The side effects of drugs and their administration require in-depth research and continuous improvement to determine the optimal therapeutic dose and ensure safety. Although currently we face numerous challenges, the caspase family is undoubtedly a promising therapeutic target for the treatment of IVDD.

## Figures and Tables

**Figure 1 biomolecules-12-01074-f001:**
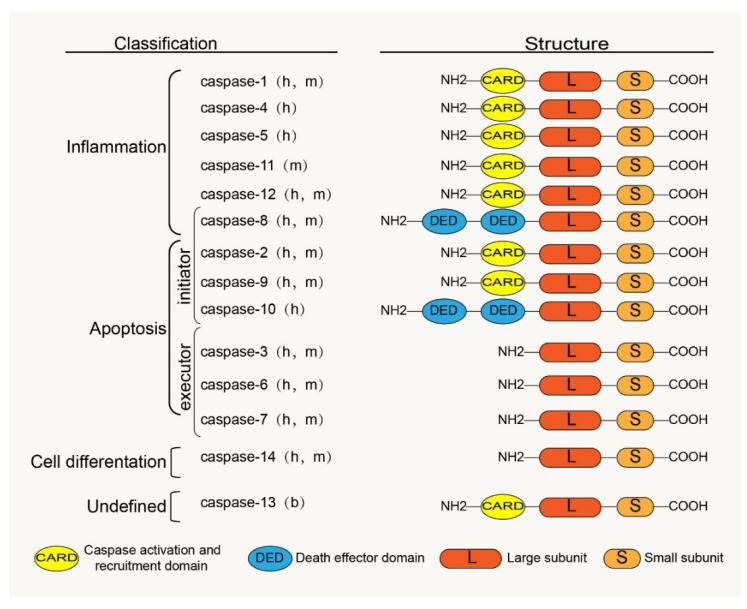
Classification, functions and the basic structure of caspase family: h, human; m, mouse; b, bovine.

**Figure 2 biomolecules-12-01074-f002:**
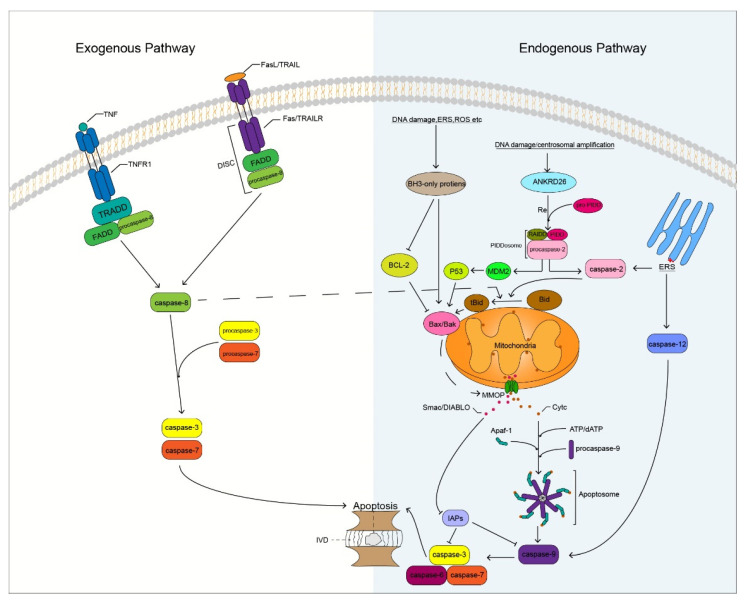
Potential mechanism of caspase involved in apoptosis of IVD cells. DISC, Death-inducing signaling complex; IVD, Intervertebral disc; ERS, Endoplasmic reticulum stress; ROS, Reactive oxygen species; MMOP, Mitochondrial outer membrane permeabilization; RE, Recruit.

**Figure 3 biomolecules-12-01074-f003:**
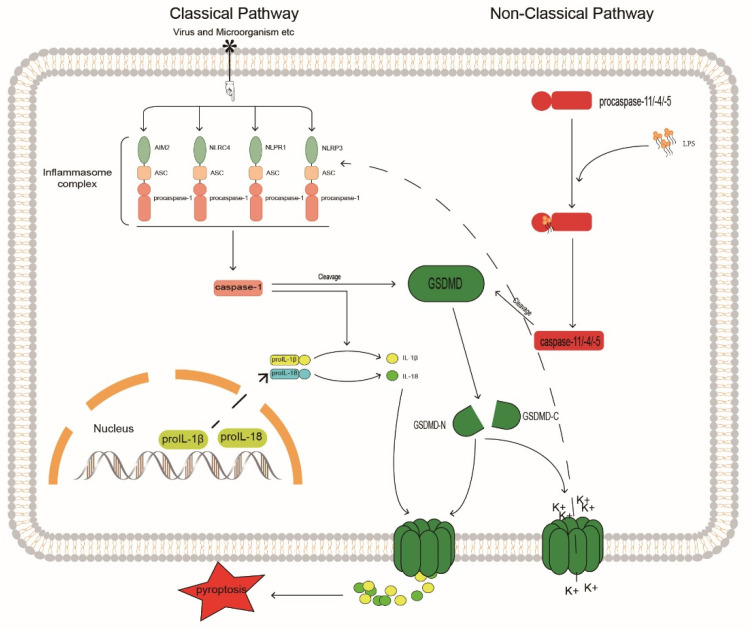
Caspase may be involved in the mechanism of inflammatory response in IVDD. GSDMD, Gasdermin D; GASDMD-N, Gasdermin D N terminal; GASDMD-C, Gasdermin D C terminal.

**Table 1 biomolecules-12-01074-t001:** Regulation of expression of some natural caspase inhibitors by noncoding RNAs in IVDD. IVDD, Intervertebral disc degeneration; miRNAs, microRNAs; lncRNAs, long non-coding RNAs; cicrRNAs, circular RNAs; NP, Nucleus pulposus; CEP, Cartilage endplate.

Natural Caspase Inhibitors	Non-Coding RNA(s)		Specimen	Target	Expression Level	Function	Reference
miR-222	miRNAs	miR-424-5p	NP	Bcl-2	Down	Apoptosis ↑	[91]
		miR-222	NP	Bcl-2	Down	Apoptosis ↑	[92]
		miR-195	NP	Bcl-2	Down	Apoptosis ↑	[149]
		miR-34a	CEP	Bcl-2	Down	Apoptosis ↑	[150]
		miR-143	NP	Bcl-2	Down	Apoptosis ↑	[151]
		miR-573	NP	Bax	Up	Apoptosis ↓	[93,152]
		miR-25-3p	NP	Bim	Up	Apoptosis ↓	[153]
		miR-125a	NP	Bak1	Up	Apoptosis ↓	[154]
	lncRNAs	HOTAIR	NP	miR-34a	Up	Apoptosis ↓	[155]
		SNHG6	NP	miR-101-3p	Down	Apoptosis ↑	[156]
		GAS5	NP	miR-155	Up	Apoptosis ↓	[157]
XIAP	cicrRNAs	VMA21	NP	miR-200c/XIAP	Up	Apoptosis ↓ Inflammation ↓	[134]

## Data Availability

Not applicable.

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
