# Peer review of "Role of Caspase Family in Intervertebral Disc Degeneration and Its Therapeutic Prospects"

_biomolecules, 2022, doi:10.3390/biom12081074_

Round 1

Reviewer 1 Report

I read with interest this review about  the role of caspases in IVDD. I have some important concern that should be addressed : 

1. the manuscript requires English editing. The grammar, the sentences and word selection should be improved

2. the paper is too long and difficult to read, i suggest to shorten it to be more readable

3. Aren't 172 references too much?

4. the clinical impact is described very shortly

Reviewer 2 Report

This is a thorough and extensive review on the role of caspases in the pathophysiology of intervertebral disc degeneration (IVDD). Authors have deeply gone through all the main aspects of the topic in light of the relevant literature. I believe that the manuscript fits the scientific audience of the Journal.

Some comments to be addressed:

- English language revision and polishing is advised due to occasional errors in the use of verb tenses, missing backspaces (especially in tables), lexicon and syntax, especially in the Introduction.

- Lines 31-39: As the audience of this Journal is predominantly composed of basic scientists, this section can be summarized in fewer and more concise concepts.

- Line 47: "Destruction" is not a scientifically sound term, considering the progressive and chronic nature of NP derangement typical of IVDD.

- Line 50: From a physical and biological perspective, elasticity is not a typical property of the NP.

- Line 52: "Loses its identity" is not scientifically appropriate.

- Lines 72-74: IVDD treatment is mainly divided into conservative and surgical, and options within the latter are not selected with a stepwise approach. Indeed, the type of surgery depends on the specific IVDD sequela to be treated, with rigid fusion being either the first or the last choice depending on the underlying condition. Therefore, this sentence should be rephrased.

- Line 82: "bone and joint diseases" is a too broad definition which could not apply to IVDD.

- Line 85: "PT" is probably a typo.

- Line 163: "RB" refers to "retinoblastoma" and not "retinocytoma" associated protein. 

- Line 198: What do authors mean for "degrees of apoptosis"?

- Lines 328-363: A study from Russo and colleagues showed that a murine model of type I diabetes showed signs of accelerated IDD associated with increased NP apoptosis. This study may be discussed in this section (Russo F, Ambrosio L, Ngo K, et al. The Role of Type I Diabetes in Intervertebral Disc Degeneration. Spine (Phila Pa 1976). 2019;44(17):1177-1185. doi:10.1097/BRS.0000000000003054)

- Line 574: NP cells are not chrondrocytes and thus should not be defined as such.

- Section 5: Considering the length of the manuscript, authors should refrain from describing in detail all the applications of caspase inhibitors in fields and models not strictly associated with IDD (i.e., lines 587-596; lines 617-628; lines 636-658).

Round 2

Reviewer 1 Report

well done